# Fraud-Proof Revenue Division on Subscription Platforms

Abheek Ghosh [1] [*]   Tzeh Yuan Neoh [2] [*]   Nicholas Teh [1] [*]   Giannis Tyrovolas [1] [*]

## Abstract

We study a model of subscription-based platforms where users pay a fixed fee for unlimited access to content, and creators receive a share of the revenue. Existing approaches to detecting fraud predominantly rely on machine learning methods, engaging in an ongoing arms race with bad actors. We explore revenue division mechanisms that inherently disincentivize manipulation. We formalize three types of manipulation-resistance axioms and examine which existing rules satisfy these. We show that a mechanism widely used by streaming platforms, not only fails to prevent fraud, but also makes detecting manipulation computationally intractable. We also introduce a novel rule, SCALEDUSERPROP, that satisfies all three manipulation-resistance axioms. Finally, experiments with both real-world and synthetic streaming data support SCALEDUSERPROP as a fairer alternative compared to existing rules.

## 1. Introduction

In September 2024, the FBI criminally charged a musician, Michael Smith, for orchestrating a scheme to fraudulently inflate his music streams on platforms such as Amazon Music, Apple Music, Spotify, and YouTube Music—and according to court documents, walked away with over US$10 million in royalty payments (United States Attorney's Office, 2024). To successfully execute his scheme, he utilized hundreds of thousands of songs created using AI, and built a complicated network of over a thousand bot accounts that artificially boost streams across these platforms billions of times. Although each stream originated from a bona fide, fee-paying account, the way the platform(s) distributed subscription revenue allowed each bot to generate more in royalties than it cost to maintain its subscription.

Subscription platforms have seen significant growth in recent years, driven by the rise of internet streaming services such as Spotify, Apple Music, Netflix, etc. For instance, the annual revenue of the music streaming industry reached US$27.6 billion in 2023, with significant increases over the last ten years (International Federation of the Phonographic Industry, 2024). Under this business model, users pay a fixed subscription fee to enjoy unlimited access to all content on the platform, typically by content creators. The platform then takes a fixed revenue cut and distributes the rest to the creators based on engagement metrics (e.g., play counts or views) and/or specific agreements between creators and platforms.

Despite efforts to curb manipulation, bad actors persist, using bots and click-farms to inflate user engagement (Drott, 2020; Snickars & Mähler, 2018). This issue is so significant that major music streaming platforms like Amazon Music and Spotify have established an industry advocacy group (Music Fights Fraud Alliance, 2025) to combat such fraud, which is estimated to cost the industry US$300 million annually (Burton, 2021). Additionally, the rise of AI-generated content introduces new challenges— platforms are increasingly flooded with synthetic tracks, videos, and live streams designed to exploit engagement-driven algorithms. This AI-generated content often amplifies fraudulent listening activities, making manipulation harder to detect.

Current machine learning (ML) approaches to this problem predominantly focus on *detecting* fraudulent activity—using sophisticated algorithms ranging from anomaly detection (Esmaeilzadeh et al., 2022) to unsupervised learning (Mollaoğlu et al., 2021) and graph neural networks (Li et al., 2021). For instance, music streaming platforms such as Spotify have proprietary models that identify whether a stream is legitimate (using meta-data such as IP location, listening patterns, and other information) and issue fines if they deem too many streams to be fraudulent (Spotify, 2025).

However, as AI continues to evolve, so do the methods used by fraudsters, leading to a continuous arms race. These bad actors increasingly leverage advanced automation tools to make fraudulent activities more sophisticated and harder to detect, challenging the robustness of existing detection frameworks and driving the need for innovative, adaptive solutions (United States Attorney's Office, 2024).

[*]Alphabetical ordering    [1]University of Oxford, UK   [2]Harvard University, USA. Correspondence to: Tzeh Yuan Neoh <tzehyuan_neoh@harvard.edu>, Nicholas Teh <nicholas.teh@cs.ox.ac.uk>.

*Proceedings of the 42nd International Conference on Machine Learning*, Vancouver, Canada. PMLR 267, 2025. Copyright 2025 by the author(s).

The root of the problem stems from the way revenue is currently distributed to content creators on most subscription-based streaming platforms: "funds from the royalty pool are allocated proportionally among artists based on their respective percentages of total streams" (United States Attorney's Office, 2024)—we call this rule GLOBALPROP.

In this paper, we tackle this problem from a *mechanism design* perspective, i.e., we mathematically formalize notions of fraud in this setting and investigate the existence of revenue division mechanisms that can inherently disincentivize fraudulent behavior, thereby reducing the industry's reliance on expensive and complex fraud detection methods to combat manipulation. Moreover, if such mechanisms exist, they could complement existing ML-based approaches by providing a foundational layer of fraud resistance. These mechanisms inherently target known forms of fraud, allowing ML systems to focus on adapting to emerging, previously unseen types of fraud that may arise in the future, ensuring continuous improvement in detecting and addressing manipulation.

Additionally, many policymakers and academics have also argued against the fairness of GLOBALPROP in favor of an alternative rule—USERPROP (which directly allocates a fixed fraction of each user's subscription fee only among the creators of the content the user consumes)—from an economic (Meyn et al., 2023; Muikku, 2017), empirical (Moreau et al., 2024), theoretical (Bergantiños & Moreno-Ternero, 2025), and legal (Dimont, 2018) perspective. Motivated by these debates, we aim to address fairness considerations in our work as well.

Lastly, the primary focus of our work is on fraudulent behavior specifically related to the creation of fake users (bots) to manipulate engagement metrics. We deliberately do not address the equally prevalent issue of widespread AI-generated content on these platforms. The legal status of such content can vary, especially since some popular artists openly release their AI-generated voices as (semi-)open-source (Josan, 2024), making its permissibility platform-dependent and governed by specific rules. Nonetheless, our work provides a principled framework for studying similar challenges. As AI continues to evolve and new forms of fraudulent behavior emerge, our approach can be extended to address these issues, provided that appropriate regulatory frameworks are established to guide the platforms.

## 1.1. Our Results

In this work, we focus on designing *manipulation-resistant* mechanisms from a computational and axiomatic perspective, setting our research apart from all previous work on this model. Although we build on the standard model for subscription platforms established in prior literature, our key contribution lies in introducing several axioms that aim to capture both resistance to manipulation and maintaining fairness and analyzing these axioms with respect to multiple revenue-division mechanisms—three from existing literature and one novel mechanism that we propose.

Moreover, we challenge the current status quo rule, GLOBALPROP, by demonstrating that *detecting suspicious activity* under this rule is computationally intractable—an important finding in this context. Since *fraud detection* (and fraud in general) is highly relevant to the ML community, we believe this result will be of particular interest to researchers and practitioners in the field.

In Section 2, we establish three fundamental properties that define the space of mechanisms we consider: *anonymity*, *neutrality*, and *no free-ridership*. The first two ensure that payoffs to artists only depend on their engagement with users. In particular, mechanisms cannot distinguish between fraudulent and genuine artists or users. No free-ridership eliminates trivial cases where an artist without engagement receives a non-zero payoff. Next, we formalize three forms of resistance to strategic manipulation. *Fraud-proofness* prevents adversaries from profitably creating new fraudulent users. *Bribery-proofness* prevents profitably bribing existing users and is a strengthening of *click-fraud-proofness* as presented in Bergantiños & Moreno-Ternero (2025). Finally, *(strong) Sybil-proofness* ensures that artists cannot gain by splitting into multiple identities or merging with others. All three axioms are novel in our setting and are motivated by real-world observations. We also introduce two additional fairness axioms—*engagement monotonicity* and *Pigou-Dalton consistency*, the latter inspired by an equitability concept in welfare economics.

In Section 3, we conduct an axiomatic study (with respect to our proposed concepts) of several rules proposed in the literature so far. Notably, we show that GLOBALPROP fails to satisfy fraud-proofness and bribery-proofness, in contrast to the other two contenders—USERPROP and USEREQ. Contributing to existing critiques of GLOBALPROP, we establish a case against GLOBALPROP through a computational lens, and in the context of fraud detection. We show that if a platform uses GLOBALPROP, detecting potentially fraudulent activity is NP-hard. We then analyze the two other existing rules: USERPROP and USEREQ. We study their axiomatic properties and prove that they satisfy our manipulation-resistance axioms, unlike GLOBALPROP. We also demonstrate that portioning rules cataloged in Elkind et al. (2023) fail all the manipulation-resistance axioms we consider.

Finally, in Section 4, we propose and study a new rule—SCALEDUSERPROP. We show that it has the same axiomatic guarantees as USERPROP but is fairer when measured by the popular "pay-per-stream" metric. We use this to quantify *maximum envy* in this setting and empirically

verify this against existing rules in Section 5.

All omitted proofs can be found in the full version of this paper.

## 1.2. Related Work

Our work considers the model proposed and studied by several recent works on (music) streaming platforms.[1]

Alaei et al. (2022) and Lei (2023) focused on a comparative study between GLOBALPROP and USERPROP. More specifically, Alaei et al. (2022) focused on providing characterizations of both rules with respect to newly proposed axioms. They were also concerned with which of these two rules could sustain a set of artists' profitability on the platform, as well as comparing them from both the platform's and the artists' perspectives. Lei (2023) pointed out the shortcomings of USERPROP. They compared the two rules primarily in terms of *egalitarian fairness* (i.e., the lowest payout among all artists) and *efficiency* (i.e., "dominance on quality profile"), but they allow for artists to vary stream quality and thus this concept is not relevant in our model.

Bergantiños & Moreno-Ternero (2025) go beyond previous works to consider a family of rules that interpolates between GLOBALPROP and USERPROP, and they provide further characterizations for both rules and their interpolation. Subsequently, Bergantiños & Moreno-Ternero (2024) introduced the Shapley index as a rule for this setting and characterized it using existing and new axioms.

Deng et al. (2024) investigate revenue-sharing mechanisms for AI-generated music platforms. Their work centers on the challenge of attributing a new, AI-created track to specific copyrighted recordings in the training data—an attribution problem that underpins royalty allocation in that setting. This challenge is fundamentally distinct from the problems we address.

A related stream of work is the *museum pass problem*, popular in the in the economics literature, and was first introduced by Ginsburgh & Zang (2001; 2003). The problem studies the sharing of revenue among museums from the sale of museum passes for a price below the aggregate admission fee of individual member museums (i.e., bundled pricing). Béal & Solal (2010) and Ginsburgh & Zang (2001; 2003) studied the problem as a *coalitional game*, whereas Casas-Méndez et al. (2011) and Estévez-Fernández et al. (2012) studied the problem as a *bankruptcy game*. Wang (2011) studied the dual version of the problem—the museum cost sharing problem. All of the works above (including several more recent works which look at the *Shapley value* as a rule (Bergantiños & Moreno-Ternero, 2015; 2016)) es-

sentially conduct an axiomatic study of popular rules in their respective games modeled, but adapted to this new setting. We refer the reader to the Casas-Méndez et al. (2014) for a survey on earlier works on this area. From 2001 to 2014, works on the topic cumulatively studied more than 30 axioms, with broadly two kinds of manipulation-resistant axioms—one based on "ticket prices" and the other based on "reported number of visitors". However, we note that the museum pass problem is fundamentally different from our problem, and thus the way axioms (and rules) are conceptualized would also naturally be distinct. This distinction is particularly apparent when it comes to concepts relating to manipulation.

Our work also contributes to the broader literature on applying computational and algorithmic methods to address incentive-related challenges in online economic systems and platforms. For example, manipulation issues have been studied in the contexts of online advertising markets (Golrezaei et al., 2021a; Kanoria & Nazerzadeh, 2014), recommendation systems (Eilat & Rosenfeld, 2023; Yao et al., 2023), and e-commerce platforms (Golrezaei et al., 2021b; He et al., 2022; Mayzlin et al., 2014).

## 2. Model and Axioms

For each positive integer $k$, let $[k] := \{1, \ldots, k\}$. Let $N = [n]$ be the set of *users* and $C = [m]$ be the set of *artists*. Suppose that an *adversary* controls a set of *fake users* $\widehat{N} \subseteq N$ and a set of *fake artists* $\widehat{C} \subseteq C$; let $\widehat{n} = |\widehat{N}|$. For each $i \in N$ and $j \in C$, let $w_{ij} \geq 0$ denote the number of *interactions* user $i$ has with artist $j$.[2] For each user $i \in N$, we assume that $\sum_{j \in C} w_{ij} > 0$, i.e., the user has some non-zero interactions.[3] Let $\mathbf{w}_i = (w_{i1}, \ldots, w_{im})$ for each $i \in N$. The *engagement profile* is $\mathbf{w} = (\mathbf{w}_1, \ldots, \mathbf{w}_n)$.

Without loss of generality, we assume that the subscription fee for each user is 1 unit. Then, the total subscription fee collected from the users is $n$. As assumed in the prior works on this topic, and as observed in the real-world, we assume that the platform takes a cut of $(1 - \alpha)n$ and distributes the remaining $\alpha n$ to the artists, for some $\alpha \in (0, 1]$.

A problem *instance* $\mathcal{I} = (N, C, \mathbf{w})$ is defined by the set of users $N$, the set of artists $C$, and the engagement profile $\mathbf{w}$. A *payment rule* (or simply *rule*) is a function $\phi$ that maps each instance $\mathcal{I}$ to an $m$-valued vector $(\phi_{\mathcal{I}}(1), \ldots, \phi_{\mathcal{I}}(m))$, where $\phi_{\mathcal{I}}(j)$ is the payment to artist $j \in C$. To simplify notation, for a subset of artists $S \subseteq C$, we use $\phi_{\mathcal{I}}(S)$ to

---

[1]However, we note that this model is also applicable to many other content subscription platforms (e.g., education, art, etc.).

[2]This is typically defined as a *stream* (on music streaming platforms like Spotify), whereby a user plays a track for a minimum duration, or a *view* (on video streaming platforms like YouTube Live) when a user joins and stays for a minimum amount of time.

[3]Note that in many of our proofs, we can without loss of generality assume that weights are rational numbers.

denote the sum of the payments to the artists in the set $S$: $\phi_{\mathcal{I}}(S) = \sum_{j \in S} \phi_{\mathcal{I}}(j)$.

## 2.1. Preliminary Axioms

We begin by introducing three fundamental fairness properties that any reasonable revenue division mechanism in our setting should satisfy. We will then provide a rationale for the necessity of these axioms in our setting.

The first axiom—anonymity—prescribes that the rule cannot distinguish between real and fake users.

**Definition 2.1** (Anonymity). A rule $\phi$ is *anonymous* if permuting the labels of the users does not affect the payoffs of the artists. Formally, rule $\phi$ is anonymous if for all instances $\mathcal{I} = (N, C, \mathbf{w})$ and $\mathcal{I}' = (N, C, \mathbf{w}')$ and all permutations $\sigma : N \to N$, if $\mathbf{w}_i = \mathbf{w}'_{\sigma(i)}$ for all users $i \in N$, then for all artists $j \in C$, $\phi_{\mathcal{I}}(j) = \phi_{\mathcal{I}'}(j)$.

The second axiom—neutrality—is similar in nature to anonymity, but for artists. It prescribes that the rule cannot distinguish between real and fake artists.

**Definition 2.2** (Neutrality). A rule $\phi$ is *neutral* if permuting the labels of the artists permutes their payoffs. Formally, rule $\phi$ is neutral if for all instances $\mathcal{I} = (N, C, \mathbf{w})$ and $\mathcal{I}' = (N, C, \mathbf{w}')$ and all permutations $\sigma : C \to C$, if $w_{ij} = w'_{i\sigma(j)}$ for all users $i \in N$ and artists $j \in C$, then for all artists $j \in C$, $\phi_{\mathcal{I}}(j) = \phi_{\mathcal{I}'}(\sigma(j))$.

In our setting, it is crucial to consider only rules that are anonymous and neutral. In practice, given the number of users/artists, it is virtually impossible to detect all fake users/artists, even with existing fraud detection techniques, as noted in our introduction. This inability to reliably distinguish between real and fake users or artists underscores the importance of addressing the questions we aim to answer.

Finally, the last fundamental axiom we consider is the notion of *no free-ridership*. Intuitively, this means that artists who receive no user engagement should not receive any payment.

**Definition 2.3** (No free-ridership). A rule $\phi$ satisfies *no free-ridership* if, for any instance $\mathcal{I} = (N, C, \mathbf{w})$ and artist $j \in C$ where $\sum_{i \in N} w_{ij} = 0$, then $\phi_{\mathcal{I}}(j) = 0$.

This axiom rules out trivial rules that allocate payments disregarding user engagement (e.g., giving equal payment to each artist irrespective of user engagement) and are, therefore, resistant to strategic manipulation.

## 2.2. Axioms for Preventing Strategic Manipulation

We start by formalizing the fraud alleged in the indictment mentioned in the introduction. Intuitively, no adversary should be able to create fake users ($\widehat{N}$), pay their subscrip-

tion fee, and earn a profit from her own fake artists ($\widehat{C}$).[4] Rules that make such fraud impossible are *fraud-proof*.

**Definition 2.4** (Fraud-proofness). A rule $\phi$ is *fraud-proof* if the following holds: For any two instances $\mathcal{I} = (N \setminus \widehat{N}, C, \mathbf{w})$ and $\mathcal{I}' = (N, C, \mathbf{w}')$ with $\mathbf{w}_i = \mathbf{w}'_i$ for all $i \in N \setminus \widehat{N}$, and any $\widehat{C} \subseteq C$, we have that $\phi_{\mathcal{I}'}(\widehat{C}) - \phi_{\mathcal{I}}(\widehat{C}) \leq \widehat{n}$.

A rule $\phi$ is *single-user fraud-proof* if $\widehat{n} = 1$.

Our definition of fraud-proofness considers only an adversary's profit from creating fake users, not fake artists. This means an adversary can introduce fake artists to earn profits without using fake users. However, without fake users, any fake artist must attract engagement from real users to profit (by the no free-ridership assumption). Whether this practice violates a platform's rules is a separate issue beyond our scope—we focus on the extra profit an adversary can gain by adding fake users, assuming a fixed set of artists (which may include fake ones).

Next, we show that single-user fraud-proofness is equivalent to (multi-user) fraud-proofness, simplifying how one can reason about fraud-proofness.

**Proposition 2.5.** *A rule $\phi$ is fraud-proof if and only if it is single-user fraud-proof.*

Another form of manipulation is *bribery*. Bribery is particularly relevant in scenarios where the platform imposes substantially stringent access requirements, making creating fake users significantly more challenging. However, under such conditions, artists may resort to colluding with and *bribing* users—offering to pay the subscription fees of the users to manipulate their engagement profiles. This practice is commonly observed in *streaming farms*, the streaming equivalent of *click farms* in advertising (Drott, 2020). We call resistance to such bribery as *bribery-proofness*.

**Definition 2.6** (Bribery-proofness). A rule $\phi$ is *bribery-proof* if the following holds: For any two instances $\mathcal{I} = (N, C, \mathbf{w})$ and $\mathcal{I}' = (N, C, \mathbf{w}')$ with $\mathbf{w}_i \neq \mathbf{w}'_i$ for exactly $k$ users, and any $\widehat{C} \subseteq C$, we have that $\phi_{\mathcal{I}'}(\widehat{C}) - \phi_{\mathcal{I}}(\widehat{C}) \leq k$.

A rule $\phi$ is *single-user bribery-proof* if $k = 1$.

Similarly to fraud-proofness, multi-user bribery-proofness and single-user bribery-proofness are equivalent.

**Proposition 2.7.** *A rule is bribery-proof if and only if it is single-user bribery-proof.*

We note that (single-user) bribery-proofness substantially strengthens the axiom of *click-fraud-proofness* proposed in Bergantiños & Moreno-Ternero (2025). Click-fraud-proofness requires that a single user altering their engagement cannot alter the payoff of any artist by more than

---

[4]Note that we do not impose any constraints on the listening behavior or engagement profiles of these fake users.

1. Formally, for all $j$, $|\phi_{\mathcal{I}'}(j) - \phi_{\mathcal{I}}(j)| \leq 1$. Single-user bribery-proofness requires that for all subsets of artists $\widehat{C} \subseteq C$, $|\phi_{\mathcal{I}'}(\widehat{C}) - \phi_{\mathcal{I}}(\widehat{C})| \leq 1$.[5] Bribery-proofness implies click-fraud-proofness and protects from multiple artists colluding.

Fraud-proofness and bribery-proofness capture resilience to two different kinds of manipulation. Despite being similar, we show that the axioms are not equivalent. Recall that $\alpha$ is the fraction of each user's subscription fee that is allocated to the artists, with the remaining portion retained by the platform as a fixed cut.

**Theorem 2.8.** *Consider some rule $\phi$. Then:*

   *(i) If $\alpha = 1$ and $\phi$ is fraud-proof, it is also bribery-proof;*

   *(ii) For $\alpha \in (0, 1]$, there exists a rule that is bribery-proof but not fraud-proof, even when $m = 2$;*

   *(iii) For $\alpha < 1$, there exists a rule that is fraud-proof but not bribery-proof, even when $m = 2$.*

The last pair of axioms that we consider—*Sybil-proofness*[6] and its strong counterpart—addresses a different form of strategic manipulation compared to the two earlier concepts. Intuitively, these axioms are designed to prevent any artist(s) from splitting or merging to gain an unfair advantage and fraudulently increasing their revenue share, thus also ensuring that all artists are treated fairly based on their actual level of user engagement.

**Definition 2.9** (Sybil-proofness)**.** A rule $\phi$ is *Sybil-proof* if the following holds: For any two instances $\mathcal{I} = (N, C, \mathbf{w})$ and $\mathcal{I}' = (N, C', \mathbf{w}')$ whereby $C \subseteq C'$, if for every subset of artists $C^* \subseteq C$ such that

   (i) $w_{ij} = w'_{ij}$ for all $i \in N, j \in C^*$; and

   (ii) $\sum_{j \in C \setminus C^*} w_{ij} = \sum_{j \in C' \setminus C^*} w'_{ij}$ for all $i \in N$,

then we must have that $\phi_{\mathcal{I}}(C \setminus C^*) = \phi_{\mathcal{I}'}(C' \setminus C^*)$.

We can define a stronger notion of Sybil-proofness by relaxing (i) and (ii), defined as follows. Note that strong Sybil-proofness implies Sybil-proofness.

**Definition 2.10** (Strong Sybil-proofness)**.** A rule $\phi$ is *strongly Sybil-proof* if the following holds: For any two instances $\mathcal{I} = (N, C, \mathbf{w})$ and $\mathcal{I}' = (N, C', \mathbf{w}')$ whereby $C \subseteq C'$, if for any subset of artists $C^* \subseteq C$ such that

   (i) $\sum_{i \in N} w_{ij} = \sum_{i \in N} w'_{ij}$ for all $j \in C^*$; and

---

[5] Note that by Proposition 2.7, it suffices to only consider single-user bribery-proofness.

[6] The name is inspired by the concept of a *Sybil attack* in computer networks.

   (ii) $\sum_{i \in N} \sum_{j \in C \setminus C^*} w_{ij} = \sum_{i \in N} \sum_{j \in C' \setminus C^*} w'_{ij}$,

then we must have that $\phi_{\mathcal{I}}(C \setminus C^*) = \phi_{\mathcal{I}'}(C' \setminus C^*)$.

We will show later that GLOBALPROP is the only neutral rule satisfying strong Sybil-proofness (Theorem 3.2), hence also motivating our study of (the weaker) Sybil-proofness.

### 2.3. Fairness Axioms

Next, we consider two fairness properties—engagement monotonicity and Pigou-Dalton consistency.

Intuitively, if an artist's engagement increases while every other artists' engagement does not increase, this artist's payoff should not decrease—this aligns with basic economic principles. It would be fundamentally unfair for a creator's rising popularity to result in a lower payoff. We formalize this fairness property as follows.

**Definition 2.11** (Engagement monotonicity)**.** A rule $\phi$ is *engagement monotone* if the following holds: For any two instances $\mathcal{I} = (N, C, \mathbf{w})$ and $\mathcal{I}' = (N, C, \mathbf{w}')$, if there exists a $j^* \in C$ such that

   (i) $w_{ij^*} \leq w'_{ij^*}$ for all $i \in N$; and

   (ii) $w_{ij} \geq w'_{ij}$ for all $i \in N$ and $j \in C \setminus \{j^*\}$,

then we must have that $\phi_{\mathcal{I}}(j^*) \leq \phi_{\mathcal{I}'}(j^*)$.

Next, the *Pigou-Dalton principle* (Pigou, 1920; Dalton, 1920), is a fundamental fairness notion from welfare economics and often referenced in collective decision-making (Moulin, 2003)—it states that among similar outcomes, the equitable one should be picked. We interpret this principle in our setting: all other things being equal, an artist who is more "uniformly enjoyed" should receive weakly more payoff from an equally popular but "polarizing" artist.

**Definition 2.12** (Pigou-Dalton consistency)**.** A rule $\phi$ is *Pigou-Dalton consistent* if the following holds: For any two instances $\mathcal{I} = (N, C, \mathbf{w})$ and $\mathcal{I}' = (N, C, \mathbf{w}')$, if there exists some $i, i' \in N$ and $j \in C$ such that

   (i) $w'_{ij} = w_{ij} - \delta$ (where $\delta > 0$ and $w_{ij} - \delta > 0$);

   (ii) $w'_{i'j} = w_{i'j} + \delta$ and $w'_{i'j} \leq w'_{ij}$; and

   (iii) $w_{kj'} = w'_{kj'}$ for all $k \in N$ and $j' \in C \setminus \{j\}$, and $w_{kj} = w'_{kj}$ for all $k \in N \setminus \{i, i'\}$.

then we must have that $\phi_{\mathcal{I}}(j) \leq \phi_{\mathcal{I}'}(j)$.

## 3. Existing Mechanisms

In this section, we formally define the three existing mechanisms proposed in the literature, and study which axioms

| Axioms / Rules | GP | UP | UEQ | ScUP |
|---|---|---|---|---|
| Fraud-proofness | ✗ | ✓ | ✓ | ✓ |
| Bribery-proofness | ✗ | ✓ | ✓ | ✓ |
| Sybil-proofness | ✓ | ✓ | ✗ | ✓ |
| Strong Sybil-proofness | ✓ | ✗ | ✗ | ✗ |
| Engagement monotonicity | ✓ | ✓ | ✓ | ✓ |
| Pigou-Dalton consistency | ✓ | ✗ | ✓ | ✗ |

*Table 1.* Axiomatic properties of the revenue division mechanisms. GP is GLOBALPROP, UP is USERPROP, UEQ is USEREQ, and ScUP is SCALEDUSERPROP.

they satisfy. We summarize our results in Table 1. At the end of the section, we also include a reference to a discussion on how our model generalizes *portioning* rules.

The rules we consider in this and the next section trivially satisfy anonymity and neutrality. Therefore, among the three preliminary axioms introduced in Section 2.1, we will only formally prove the satisfaction of no free-ridership.

### 3.1. GLOBALPROP: The Status Quo

GLOBALPROP distributes the payoff to each artist proportionally to the artist's share of total engagement. For example, if there are $500$ users, and an artist gets $25\%$ of the total user engagement in the platform, then the artist correspondingly receives a payment of $0.25 \times 500\alpha = 125\alpha$ under GLOBALPROP. According to court documents (United States Attorney's Office, 2024), this is the rule that major streaming platforms use.[7]

---

**GLOBALPROP**

Given an instance $\mathcal{I} = (N, C, \mathbf{w})$ and for each $j \in C$, the payment rule GLOBALPROP is defined as follows.

- - - - - - - - - - - - - - - - - - - - - - - - - - - - -

$$\phi_{\mathcal{I}}(j) = \frac{\sum_{i \in N} w_{ij}}{\sum_{j' \in C} \sum_{i \in N} w_{ij'}} \times \alpha n.$$

---

It is easy to observe that users with higher engagement exert a disproportionate influence on revenue distribution. Given this, it is not surprising that this rule fails to satisfy both fraud-proofness and bribery-proofness.

**Theorem 3.1.** GLOBALPROP *satisfies strong Sybil-proofness, but fails fraud-proofness and bribery-proofness.*

Moreover, strong Sybil-proofness uniquely characterizes GLOBALPROP, given our neutrality assumption.

**Theorem 3.2.** GLOBALPROP *is the only neutral rule satisfying strong Sybil-proofness.*

GLOBALPROP also satisfies our fairness axioms.

---

[7]It is also sometimes known as the *pro-rata* rule.

**Theorem 3.3.** GLOBALPROP *satisfies no free-ridership, engagement monotonicity, and Pigou-Dalton consistency.*

**A Case Against GLOBALPROP: The Computational Intractability of Fraud Detection.** We have shown that GLOBALPROP is not fraud-proof. One might hope that artists benefiting from fraud could be easily identified and removed. Unfortunately, detecting the artists who gain the most from fraudulent activity is computationally intractable.

Importantly, a user who streams music extensively is not inherently suspicious—some people naturally listen to music for most of their waking hours. Thus, instead of targeting individual active users, we should focus on identifying artists who may be used as vehicles for fraud by an adversary.[8]

**Definition 3.4** (Potentially Suspicious Profits). Given a set of artists $U \subseteq C$, their *potentially suspicious profit (*PSP*)* from GLOBALPROP is their maximum marginal profits from a set of users $V$, less the cost of creating these users:

$$\mathrm{PSP}(U) = \max_{V \subseteq N} \left( \frac{\sum_{i \in N} \sum_{j \in U} w_{ij}}{\sum_{i \in N} \sum_{j \in C} w_{ij}} \times \alpha n \right.$$
$$\left. - \frac{\sum_{i \in N \setminus V} \sum_{j \in U} w_{ij}}{\sum_{i \in N \setminus V} \sum_{j \in C} w_{ij}} \times \alpha(n - |V|) - |V| \right).$$

Thus, our objective of identifying suspicious artists can be framed as finding a set of artists $U \subseteq C$ such that $\mathrm{PSP}(U)$ is high. However, the choice of $|U|$ is crucial. If we restrict $U$ to a single artist ($|U| = 1$), an adversary can easily evade detection by distributing fake users' listening activity across multiple fraudulent artists. On the other hand, if we impose no constraint on $|U|$, we risk identifying a set of legitimate artists with dedicated fan bases. Also, while an adversary can create multiple fake artists, doing so incurs administrative overhead—such as setting up identification and banking details—which makes the creation of an arbitrarily large number of fake artists impractical in many circumstances.

Therefore, we define the problem of finding suspicious artists as finding the set $U \subseteq C$ of size at most $k$ artists that maximize $\mathrm{PSP}(U)$. However, we show that this problem is computationally intractable, with the following result.

**Theorem 3.5.** *Given an instance $\mathcal{I} = (N, C, \mathbf{w})$ and parameters $k \leq |C|$ and $\gamma > 0$, it is NP-hard to determine if there exists a $U \subseteq C$ such that $|U| \leq k$ and $\mathrm{PSP}(U) \geq \gamma$.*

### 3.2. User-Additive Rules

At the opposite extreme from GLOBALPROP are rules where each user's subscription fee is distributed solely based on their individual engagement profile. Under these rules, an

---

[8]Our objective is to identify fraudulent artists as a means of detecting suspicious interactions between fake users and fake artists.

artist's total payoff is simply the sum of the amounts they would receive from each user in a single-user setting. We refer to this class of rules as *user-additive*.[9]

**Definition 3.6** (User-additive rules). For each instance $\mathcal{I} = (N, C, \mathbf{w})$, define instances $\mathcal{I}_i = (\{i\}, C, \mathbf{w}_i)$ for each $i \in N$. Then, a rule $\phi$ is *user-additive* if for all instances $\mathcal{I}$ and artists $j \in C$, $\phi_{\mathcal{I}}(j) = \sum_{i \in N} \phi_{\mathcal{I}_i}(j)$.

We then show the following.

**Proposition 3.7.** *A user-additive rule is fraud-proof and bribery-proof.*

We focus on two user-additive rules that have been discussed in the existing literature: USERPROP and USEREQ. Under USERPROP, an $\alpha$ fraction of each user's subscription fee is allocated to the artists proportional to the user's engagement. For example, if a user listens to three artists—the first artist $50\%$ of the time and the other two artists $25\%$ each—then under USERPROP, the artists will receive payments of $\alpha/2$, $\alpha/4$, and $\alpha/4$ from this user, respectively. The total payment of an artist is the sum of such payments from each user.

---

**USERPROP**

Given an instance $\mathcal{I} = (N, C, \mathbf{w})$ and for each $j \in C$, the payment rule USERPROP is defined as follows.

- - - - - - - - - - - - - - - - - - - - - - - - - - - -

$$\phi_{\mathcal{I}}(j) = \sum_{i \in N} \frac{w_{ij}}{\sum_{j' \in C} w_{ij'}} \times \alpha.$$

---

We show that it satisfies all of the manipulation-resistant axioms (excluding strong Sybil-proofness) and engagement monotonicity, but fails Pigou-Dalton consistency.

**Theorem 3.8.** USERPROP *is fraud-proof, bribery-proof, and Sybil-proof, but fails strong Sybil-proofness.*

**Theorem 3.9.** USERPROP *satisfies no free-ridership and engagement monotonicity, but fails Pigou-Dalton consistency.*

Next, we consider the USEREQ rule, first studied in Bergantiños & Moreno-Ternero (2024). They established the equivalence between USEREQ and the *Shapley value*, a fundamental measure in cooperative game theory that ensures a fair distribution of payoffs among players based on their contributions (Shapley, 1953).

Now, given an instance $\mathcal{I} = (N, C, \mathbf{w})$, for each $i \in N$ and $j \in C$, let $\mathbf{1}_{w_{ij}>0}$ be the indicator function that returns the value 1 if $w_{ij} > 0$, and 0 otherwise. In USEREQ, an $\alpha$ fraction of each user's subscription fee is distributed equally among the artists with strictly positive engagement from the user. For example, if a user listens to only three artists—$80\%$, $19\%$, and $1\%$ of the time, respectively—and does not

listen to other artists, then these three artists each receives a payment of $\alpha/3$ from this user, and the remaining artists do not receive any payment from the user. The total payment to an artist is the sum of such payments from each user.

---

**USEREQ**

Given an instance $\mathcal{I} = (N, C, \mathbf{w})$ and for each $j \in C$, the payment rule USEREQ is defined as follows.

- - - - - - - - - - - - - - - - - - - - - - - - - - - -

$$\phi_{\mathcal{I}}(j) = \sum_{i \in N} \frac{\mathbf{1}_{w_{ij}>0}}{|\{j' \in C : w_{ij'} > 0\}|} \times \alpha.$$

---

USEREQ has similar guarantees as USERPROP, with the difference being that it fails Sybil-proofness, but satisfies Pigou-Dalton consistency.

**Theorem 3.10.** USEREQ *is fraud-proof and bribery-proof, but fails Sybil-proofness.*

**Theorem 3.11.** USEREQ *satisfies no free-ridership, engagement monotonicity, and Pigou-Dalton consistency.*

**A Generalization of Portioning**

We also make an important observation: our model can be viewed as a generalization of *portioning* under cardinal preferences (Elkind et al., 2023; Freeman et al., 2021), where each agent subjectively divides a contiguous resource (such as time or money) among a given set of *candidates*, and the goal is to aggregate these preferences to obtain one (fair) division. This is similar to our model if we let agents be users, candidates be artists, and preferences be interactions.[10] However, portioning rules require that the engagement of each user is normalized (i.e., sums to 1). We can then generate rules for our setting by normalizing each $\mathbf{w}_i$ and applying a portioning rule to the instance. There are eight portioning rules cataloged in Elkind et al. (2023). One of them is equivalent to USERPROP, but the other seven fail fraud-proofness, bribery-proofness and Sybil-proofness. We present these rules and prove the results in the full version of this paper.

## 4. SCALEDUSERPROP: A Fairer Mechanism

The three rules we considered above are conceptually distinct: GLOBALPROP allows dedicated fans to exert a disproportionate influence on revenue distribution, but this also creates opportunities for fraud by fabricating users who may *appear* as dedicated fans. In contrast, USERPROP is often viewed by policymakers as a more desirable alternative to GLOBALPROP. However, USERPROP is not necessarily fairer (Lei, 2023), and user-additive rules in general may fail to meaningfully reward artists for increasing the engage-

---

[9]This term is distinct from *user-centric*, which is sometimes used in the literature to refer to USERPROP.

[10]We note that this analogy requires imposing rational number constraints on preferences, as assumed in the preliminaries.

ment within their existing fanbase.

To better understand differences in *payment fairness*, it is useful to examine the *pay-per-stream* metric (Dimont, 2018; Meyn et al., 2023). Given an instance $\mathcal{I}$ and an artist $j$, let the artist *pay-per-stream* (PPS) for rule $\phi$ be $\text{PPS}(\phi, \mathcal{I}, j) = \frac{\phi_{\mathcal{I}}(j)}{\sum_{i \in N} w_{ij}}$. Using this, we define the *maximum envy* (ME) of $\mathcal{I}$ as $\text{ME}(\phi, \mathcal{I}) = \frac{\max_{j \in C} \text{PPS}(\phi, \mathcal{I}, j)}{\min_{j' \in C} \text{PPS}(\phi, \mathcal{I}, j')}$. This ratio quantifies the disparity in PPS between the highest-paid and lowest-paid artists, providing a measure of the maximum envy in revenue distribution.

Then, we obtain the following result, which essentially implies that any fraud-proof or bribery-proof rule has the potential to be extremely unfair (unbounded maximum envy).

**Proposition 4.1.** *For all* $\alpha \in (0, 1]$ *and rules* $\phi$, *if there exists* $k \in \mathbb{R}$ *such that for all instances* $\mathcal{I}$, $\text{ME}(\phi, \mathcal{I}) \leq k$, *then* $\phi$ *fails fraud-proofness and bribery-proofness.*

However, not all such rules may perform equally bad on this front—we will analyze this later through experiments (in Section 5), with a slight variant of the ME definition.

Given this, we attempt to achieve a compromise by designing a rule that has the same axiomatic guarantees as USERPROP, while offering empirically (in Section 5) stronger fairness guarantees than USERPROP and USEREQ. SCALEDUSERPROP works by having the platform take a *disproportionate amount* of commission from low-engagement users. The platform then runs USERPROP on the remaining subscription fees. It is defined as follows.

---

SCALEDUSERPROP

Given an instance $\mathcal{I} = (N, C, \mathbf{w})$, let $\gamma$ be a constant such that $\sum_{i \in N} \min\left(\gamma \cdot \sum_{j \in C} w_{ij}, 1\right) = \alpha n$. Then, for each $j \in C$, the payment rule SCALEDUSERPROP is defined as follows.

$$\phi_{\mathcal{I}}(j) = \sum_{i \in N} \left( \min(\gamma \cdot \sum_{j' \in C} w_{ij'}, 1) \times \frac{w_{ij}}{\sum_{j' \in C} w_{ij'}} \right).$$

---

Note that when $\alpha = 1$, we have $\min(\gamma \cdot \sum_{j' \in C} w_{ij'}, 1) = 1$ for all $i \in N$, making SCALEDUSERPROP equivalent to USERPROP. For $\alpha < 1$, if no user's engagement exceeds $\frac{1}{\alpha}$ times the average engagement, then SCALEDUSERPROP is equivalent to GLOBALPROP, which we show below.

**Theorem 4.2.** *Fix an instance* $\mathcal{I} = (N, C, \mathbf{w})$. *If* $\sum_{j \in C} w_{ij} \leq \frac{1}{n\alpha} \sum_{i \in N} \sum_{j \in C} w_{ij}$ *for all* $i \in N$, *then* SCALEDUSERPROP *is equivalent to* GLOBALPROP.

Thus, SCALEDUSERPROP can be viewed as a variant of GLOBALPROP that "limits the influence" of users who have engagement significantly above average. We then show

that SCALEDUSERPROP has exactly the same axiomatic guarantees as USERPROP, with the following results.

**Theorem 4.3.** SCALEDUSERPROP *satisfies fraud-proofness, bribery-proofness, and Sybil-proofness, but fails strong Sybil-proofness.*

**Theorem 4.4.** SCALEDUSERPROP *satisfies no free-ridership, engagement monotonicity, but fails Pigou-Dalton consistency.*

## 5. Experiments

To complement our theoretical analysis, we conduct experiments to evaluate our fraud-proof (and bribery-proof) mechanisms—USERPROP, USEREQ, SCALEDUSERPROP—using both synthetic and real-world datasets. Motivated by our definition of *maximum envy* in Proposition 4.1, for each rule, we analyze the top and bottom few users based on their *pay-per-stream (*PPS*)* relative to GLOBALPROP's PPS, as the revenue share ($\alpha$) varies.[11] Notably, only SCALEDUSERPROP is influenced non-linearly by changes in $\alpha$ (the other rules scale linearly with $\alpha$). Consequently, the pay-per-stream values for the other three rules remain constant across different values of $\alpha$.

**Synthetic datasets** We generate synthetic problem instances involving $10,000$ users and $1,000$ artists. For each user, we first determine the number of artists they interact with by drawing a value uniformly at random from the range $[1, 100]$. Based on this value, we randomly select the corresponding number of artists from the pool of $1,000$. For each chosen artist, the number of times the user streams their music is sampled from a Poisson distribution with $\lambda = 1$. We repeat the experiments $100$ times.

**Real-world datasets** We utilize data from the *Music Listening Histories Dataset* (Vigliensoni & Fujinaga, 2017), that contains the listening history of approximately $583,000$ users, $439,000$ artists, and a cumulative total of $27$ billion *listening events* (i.e., user-artist interactions).[12]

**Discussion** On real-world data, SCALEDUSERPROP emerges as fairest mechanism among those considered, especially for values of $\alpha$ not close to 1; whereas USEREQ, which treats avid and casual listeners equally, is the least fair. SCALEDUSERPROP significantly reduces the top 100

---

[11]Note that in Proposition 4.1, maximum envy is defined with respect to the single top and bottom user, which differs from the metric used in this section. In our experiments, we chose to report metrics for the top and bottom few users rather than just the single best and worst, as we believe this provides a more robust assessment—mitigating the impact of potential outliers that may disproportionately affect the extremes. However, our definition and theoretical results would easily extend to top and bottom few users, making it consistent with that used for the experiments.

[12]Our code is accessible at https://github.com/nicteh/Fraud-Proof-Revenue-Division.

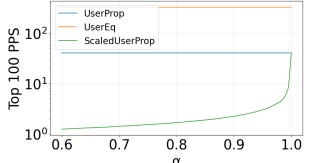

(a) Real data, top 100 artists' PPS relative to GP

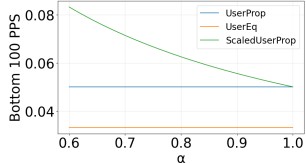

(b) Real data, bottom 100 artists' PPS relative to GP

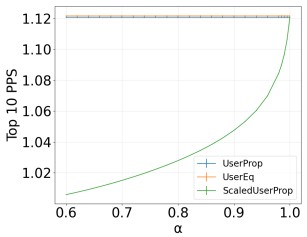

(c) Synthetic data, top 10 artists' PPS relative to GP

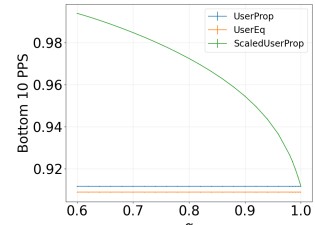

(d) Synthetic data, bottom 10 artists' PPS relative to GP

*Figure 1.* Overview of graphs from real and synthetic data. (a) and (b) show results for real data, while (c) and (d) show results for synthetic data. GP is short for GLOBALPROP.

artists' PPS even for $\alpha > 0.9$, but it only gradually increases the bottom 100 PPS as $\alpha$ decreases. To understand this outcome, we first observe that artists with high PPS typically attract infrequent listeners, while those with low PPS tend to have a more dedicated, avid fanbase.

We also observe that under SCALEDUSERPROP, each stream from a user contributes $\min(\gamma, \frac{1}{\sum_{j \in C} w_{ij}})$, whereas under USERPROP, it contributes $\frac{\alpha}{\sum_{j \in C} w_{ij}}$. For avid listeners with high $\sum_{j \in C} w_{ij}$, a stream under SCALEDUSER-PROP is worth $\frac{1}{\alpha}$ times its value under USERPROP. Conversely, for infrequent listeners, SCALEDUSERPROP caps a stream's worth at $\gamma$, while under USERPROP, it can reach up to $\alpha$ in the extreme case where $\sum_{j \in C} w_{ij} = 1$.

On synthetic data, SCALEDUSERPROP remains the fairest mechanism as $\alpha$ decreases. However, in contrast to the real-world data, we observe two key differences: (1) the top and bottom PPS are much closer in magnitude, and (2) USERPROP and USEREQ perform nearly identically. These differences can be partly attributed to the way synthetic instances are generated. While our model accounts for users with varying streaming frequencies, it does not capture the real-world tendency of certain artists to attract predominantly avid or infrequent listeners.

## 6. Conclusion

In this work, we formalized three types of manipulation by fraudulent agents in subscription-based platforms, motivated

by a real-world multi-million dollar fraud case. We show that the axioms we introduced are not equivalent and study the rules that satisfy them. GLOBALPROP, which is used by streaming platforms, does not satisfy fraud-proofness or bribery-proofness. However, we show that USERPROP and USEREQ do. We introduce a novel rule, SCALEDUSER-PROP. It is as strong in resisting manipulation as USERPROP and incentivizes artists to increase their overall engagement similarly to GLOBALPROP. Our empirical study on real and synthetic data of fraud-proof rules support SCALEDUSER-PROP is a fairer fraud-proof alternative to other rules

A natural follow-up direction would be to study a *freemium* model, by incorporating users who have to watch advertisements to gain access to content on the platform, and have been adopted by streaming platforms such as YouTube and Spotify, among others. Revenue division in this context would have different considerations and call for more appropriate axioms to be defined. Machine learning approaches have been adopted here as well (Goli et al., 2024); it would be interesting to explore these questions from a mechanism design perspective.

## Impact Statement

This paper presents work whose goal is to advance the theory of machine learning. There are many potential societal consequences of our work, none which we feel must be specifically highlighted here.

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
