# OpenReview forum: "Fraud-Proof Revenue Division on Subscription Platforms"
_ICML.cc/2025/Conference — ICML 2025 poster_

### Official Review · Reviewer_PcgM · 2025-03-10

**Overall Recommendation:** 4

**Summary:**

The paper "Fraud-Proof Revenue Division on Subscription Platforms" addresses the problem of revenue distribution on subscription-based platforms, particularly in the context of music streaming services. The authors formalize three types of manipulation-resistance axioms—fraud-proofness, bribery-proofness, and Sybil-proofness—and evaluate existing revenue division mechanisms against these axioms. They find that the widely used GlobalProp mechanism fails to prevent fraud and makes fraud detection computationally intractable. The authors propose a novel mechanism, ScaleDUserProp, which satisfies all three manipulation-resistance axioms and is shown to be a fairer alternative through both theoretical analysis and empirical evaluation on real-world and synthetic datasets. The paper also introduces fairness axioms such as engagement monotonicity and Pigou-Dalton consistency, and demonstrates that ScaleDUserProp performs well on these metrics compared to existing rules like UserProp and UserEQ.

**Claims And Evidence:**

The claims made in the paper are generally well-supported by clear and convincing evidence. The authors provide rigorous theoretical proofs for their proposed axioms and mechanisms, and they back up their claims with empirical experiments on both synthetic and real-world datasets. The theoretical results are well-articulated, and the empirical results are presented with sufficient detail, including the use of the Music Listening Histories Dataset, which adds credibility to their findings.

**Essential References Not Discussed:**

The paper does a good job of citing relevant prior work. I am not sure if they cited all key works since I am not familiar with this topic.

**Experimental Designs Or Analyses:**

The experimental designs and analyses are sound. The authors conduct experiments on both synthetic and real-world datasets, and they provide a clear explanation of their methodology. The results are presented in a way that allows for easy comparison between different mechanisms, and the authors discuss the implications of their findings in detail. The use of synthetic data helps to control for certain variables, while the real-world data provides a realistic evaluation of the mechanisms.

**Methods And Evaluation Criteria:**

The proposed methods and evaluation criteria are appropriate for the problem at hand. The authors use a combination of theoretical analysis and empirical evaluation to validate their claims. The synthetic data generation process is well-described, and the use of real-world data from the Music Listening Histories Dataset adds robustness to their findings. The evaluation metrics, such as pay-per-stream (PPS) and maximum envy (ME), are well-chosen to measure fairness and manipulation resistance.

**Other Comments Or Suggestions:**

None

**Other Strengths And Weaknesses:**

Strengths:

* The paper addresses a timely and important problem in the context of subscription-based platforms, particularly in the music streaming industry.

* The proposed ScaleDUserProp mechanism is novel and offers a compelling solution to the problem of manipulation resistance.

* The paper provides a thorough theoretical analysis, supported by empirical evidence, which adds to its credibility.

* The authors do a good job of connecting their work to the broader literature on fairness and revenue division.

Weaknesses:

* While the paper is well-written, some parts of the theoretical analysis could be more accessible to readers who are not familiar with the
technical details of revenue division mechanisms.

* The paper could benefit from a more detailed discussion of the limitations of the proposed mechanism, particularly in terms of scalability and computational complexity.

**Questions For Authors:**

1. Comparison with Other Mechanisms: The paper compares ScaleDUserProp with GlobalProp, UserProp, and UserEQ. Are there other mechanisms in the literature that the authors considered but did not include in their analysis? If so, why were they excluded?

**Relation To Broader Scientific Literature:**

The paper is well-situated within the broader scientific literature on revenue division and manipulation resistance in subscription platforms. The authors reference prior work on revenue-sharing mechanisms, such as GlobalProp and UserProp, and they build on these ideas to propose a new mechanism that addresses the limitations of existing approaches. The paper also connects to the literature on fairness axioms and cooperative game theory, particularly in the context of the Shapley value and Pigou-Dalton consistency.

**Theoretical Claims:**

The theoretical claims in the paper appear to be correct. The authors provide detailed proofs for their key results, including the manipulation-resistance properties of the proposed mechanisms (e.g., Theorems 3.1, 3.8, 4.3) and the fairness axioms (e.g., Theorems 3.3, 4.4). The proofs are well-structured and logically sound, and the authors provide additional proofs in the appendix for omitted results.

---

> ### Author Rebuttal · Authors · 2025-03-31
>
> Thank you for your review. Please find our response below.
>
> ---
> > Comparison with Other Mechanisms: The paper compares ScaledUserProp with GlobalProp, UserProp, and UserEQ. Are there other mechanisms in the literature that the authors considered but did not include in their analysis? If so, why were they excluded?
>
> To the best of our knowledge, these are the only rules that have been proposed and studied in the literature to date.

---

> > ### Comment · Reviewer_PcgM · 2025-04-01
> >
> > Thank you for your response. I'll keep my score.

---

### Official Review · Reviewer_yowW · 2025-03-11

**Overall Recommendation:** 2

**Summary:**

The authors propose a mechanism-design framework to counter manipulation in subscription-based streaming platforms. They formally define three types of fraudulent behaviors and illustrate how the commonly used “global proportion” revenue rule is highly vulnerable. To address this, they introduce some new axioms and prove that no single existing mechanism can satisfy them all. Then, they propose ScaledUserProp, which is a revenue-sharing mechanism designed to resist manipulation while incentivizing engagement. Experiments on some real-world streaming data demonstrate that ScaledUserProp reduces fraudulent activity compared to current industry practices.

**Claims And Evidence:**

Overall, I found this paper to be well written, both in their setup and methods. This does indeed seem to be an interesting and well motivated economics problem, and the authors seem to make a good case for their ScaledUserProp mechanism.

My main issue: I am not a computational economist, and did not expect to be reviewing economics papers for ICML. ICML is potentially the wrong venue for this work, and I worry that this may lead to a lower reviewing bar than at, say, Economics and Computation.

After a quick skim of the references in this paper, I found no publications at ICML / Neurips / ICLR. The closest: a paper published at KDD, and another at SODA. Most of the cited papers, on the other hand, were published at economics conferences / journals.

That said (and despite my lack of background), I did my best to review the paper for the benefit of the authors. However, I feel my lack of a sense of prior work has prevented me from asking pertinent and non-trivial questions. I’d like the AC to take this into account when using my review.

**Essential References Not Discussed:**

Hard for me to say - I am not familiar with this area.

**Experimental Designs Or Analyses:**

I did take a close look at the provided notebooks / “real world experiments.” This is probably the weakest portion of the paper. It’s hard to evaluate these plots. The behavior between the synthetic and “real” data seems analogous, although there are clear differences when interpolating alphas, and the tradeoffs are not explored empirically in the rigorous way that experiments tend to be presented at ICML, simply given for these two scenarios. More extensive/well interpreted empirical results (that ask questions through ablations, different ways of looking at the data, etc.), which help the reader through the use case and impacts on the artists, and would help for a conference like ICML.

**Methods And Evaluation Criteria:**

(discussed below)

**Other Comments Or Suggestions:**

N/A

**Other Strengths And Weaknesses:**

N/A

**Questions For Authors:**

My main question (beyond some of the clarifying questions above): can you convince me that there’s an audience for this work at ICML? Can you give me an example of a similar strain of work that's been presented at the conference in say, the past 3 years? I don't see the point in this paper being presented at the conference, in place of some other worthy paper, if it has no audience. This led me to set my score as "Weak Reject," despite the paper's clarity of presentation and interesting problem, but I am open to being shown that I am wrong!

**Relation To Broader Scientific Literature:**

Hard for me to say - I am not familiar with this area.

**Theoretical Claims:**

I did not check all of the theoretical claims, as they were extensive and significantly outside of my area. However, I did try to understand some of the claims, and follow some proofs. Overall, the claims I spent time with were reasonably stated. The proofs were mostly believable, but may have omitted some important details (hard for me to tell, again outside my area, so not sure what the norms are). Below, I offer some questions / comments.

**Bribery proofness:** So, the definition states that for any two instances where the engagement profiles differ for exactly $k$ users, $$\left|\phi_{\mathcal{I}}(\hat{C}) - \phi_{\mathcal{I}'}(\hat{C})\right| \leq 1$$ This bound of “1”' is independent of $k$. This surprised me: in other words, even if many users’ weights are changed (i.e., $k > 1$), the change in the allocated revenue for any subset $\hat{C}$ is still bounded by 1. This is quite strong? My naive take is that this mixes the single-user notion (as in click-fraud-proofness) with a multi-user scenario. Are we trying to allow only a unit change *regardless* of the number of bribed users?

**Clarity in Sybil-Proofness and Strong Sybil-Proofness:** For definition 2.9 of Sybil-proofness, there’s a statement “no artist benefits from splitting or merging,” and then in the formal definition its given that the condition “for any two instances $\mathcal{I} = (N,C,\mathbf{w})$ and $\mathcal{I}' = (N,C,\mathbf{w}')$ with $\mathbf{w}_i = \mathbf{w}'_i$ for each user that is not a Sybil identity.” I found the terms “splitting” and “merging” a little bit too informal, which made it hard for me to understand the Sybil proofness property - can you formalize these? Did I miss something?

**Question on the a “Pigou-Dalton Consistency” statement:** So at some point, you state that ScaledUserProp satisfies no free-ridership, engagement monotonicity, and Pigou-Dalton consistency. Then, later in "Proof of Theorem 4.4" and in Table 1, you explicitly state/show that ScaledUserProp **fails** Pigou-Dalton consistency for every $\alpha \in (0,1]$. So which one is it? Am I missing something?

**Proof of Theorem 3.2** I tried to follow this proof more closely. There were a few steps that seemed a little unclear to me. When you say “by linearity” and then make an assumption on the sum of $w_{ic}$, and then say $f(0,T,N) = 0$, I’m not entirely sure I see how this works out.

---

> ### Author Rebuttal · Authors · 2025-03-31
>
> Thank you for your review. Please find our responses below.
>
> ---
> > Main question: can you convince me that there’s an audience for this work at ICML? ... This led me to set my score as "Weak Reject," despite the paper's clarity of presentation and interesting problem, but I am open to being shown that I am wrong!
>
> While this specific line of work on subscription platforms is relatively new (motivated from economics), our work is part of a broader literature on using computational/algorithmic methods to tackle incentive challenges in online economic systems/platforms (these works aim to contribute theoretical foundations that complements more applied ML work). This is a central concern in the EconCS community, which has a growing presence at ICML.
>
> Some recent ICML papers on strategic manipulation/incentive compatibility on online systems/platforms:
> - Human vs. Generative AI in Content Creation Competition: Symbiosis or Conflict? ICML’24
> - How Bad is Top-K Recommendation under Competing Content Creators? ICML’23
> - Performative Recommendation: Diversifying Content via Strategic Incentives. ICML’23
>
> The type of “adversarial fraud” we study in subscription platforms also has a close analogue in recommendation systems (which in itself is actively studied in venues like ICML and NeurIPS), where similar challenges are known as poisoning attacks.
>
> Several other examples of work at ICML looking at strategic manipulation/incentive compatibility:
> - Online mechanism design for information acquisition. ICML’23
> - Fairness Interventions as (Dis)Incentives for Strategic Manipulation. ICML’22
> - Making Paper Reviewing Robust to Bid Manipulation Attacks. ICML’21
> - Incentivizing Compliance with Algorithmic Instruments. ICML’21
> - Strategyproof Mean Estimation from Multiple-Choice Questions. ICML’20
>
> Thus, we believe our paper fits naturally within this growing line of research at ICML and would garner interest from the community.
>
> Moreover, ICML'24 has hosted multiple workshops that reflect this interest. For instance:
> - “Agentic Markets” workshop focuses on the intersection of market/incentive design and agentic AI, and derives insights from “economics, mechanism design, game theory”.
> - “Humans, Algorithmic Decision-Making and Society: Modeling Interactions and Impact” workshop includes a topic on “Strategic behavior and its impact on algorithmic decision-making”.
> - “Next Generation of AI Safety” includes “Agentic AI” as a theme, where “adversary exploitation” of deployed AI systems is a concern.
> - “Models of Human Feedback for AI Alignment” workshop specifically calls for perspectives from “economics”
>
> These are all close thematic matches. Perhaps one additional relevant reference (will add this citation) is the paper “Computational Copyright: Towards A Royalty Model for Music Generative AI” at the ICML'24 Workshop on Generative AI and Law, but they focus on royalty models for AI-generated music.
>
> We hope that these examples of thematically similar papers and workshops from recent years have persuaded you that there is an audience at ICML for our work; and if so, that you would be able to reconsider your evaluation of our work.
>
> ---
> > ... Are we trying to allow only a unit change regardless of the number of bribed users?
>
> Nope. If $k$ users are bribed, then we allow at most $k$ units of change as defined in Def 2.6. When we compare bribery-proofness (BP) to click-fraud proofness, we focus on the case of “a single user altering their engagement”. Then, by BP, the difference for any subset of artists is 1. We can make this wording clearer. As a side note, by Prop 2.7 it suffices to only consider single-user BP.
>
> ---
> > Proof of Thm 3.2
>
> We consider linearity as used in linear algebra, and thus exclude affine functions (will clarify this). Formally, $f\left(\sum_{i \in N} w_{ic},\sum_{i \in N} \sum_{j \in C} w_{ij},N\right) = \sum_{i \in N} w_{ic} \times g\left(\sum_{i \in N} \sum_{j \in C} w_{ij},N\right)$. Clearly, if  $\sum_{i \in N} w_{ic} = 0$ then for all args $T, N$,  $f(0, T, N) = 0$.
>
> ---
> > ... More extensive/well interpreted empirical results (that ask questions through ablations, different ways of looking at the data, etc.), which help the reader through the use case and impacts on the artists, ...
>
> Our main contributions lie in introducing a novel approach to the problem, along with a set of axioms—particularly focused on manipulation-resistance—and supporting theoretical results. We agree that more extensive experiments would provide valuable insights. However, given the space/scope constraints of a conference paper, we prioritized highlighting what we believe are the most novel and fundamental aspects of the problem. But we appreciate the suggestion and will consider this for future work, thanks!
>
> ---
> We have noted other comments and will address them accordingly (adding an example to better understand the def of (Strong) Sybil-proofness, and fixing the typo in the statement for Pigou-Dalton Consistency).

---

> > ### Comment · Reviewer_yowW · 2025-04-01
> >
> > I appreciate the author's careful response to my review. I have also read through responses from the other reviewers.
> >
> > Thank you for providing me a laundry list of prior work. I took a look at the first few papers; they focus on optimization / algorithmic approaches to solving some objective. They each cite extensive prior work from the ML community, in particular work published at ICML / Neurips / ICLR.
> >
> > Let me say this: I believe that ICML should accept EC work. However, the point I was trying (and perhaps failed) to make was that ICML reviewers have a particular expertise and knowledge-base; namely, problems around optimization and learning algorithms.
> >
> > If a paper on EC contains an interesting application of those algorithmic tools, or if that EC work leads to the expansion of those tools, then it seems like a natural fit for ICML.
> >
> > However, if thats not the case, and its difficult to find prior work published at ICML / Neurips / ICLR that's relevant to cite (as it appears to be in your case), then maybe you're submitting to the wrong venue.
> >
> > I had hoped that other reviewers would be more expert on the topic, but as I suspected, three of us had little familiarity with the literature, and only one had some familiarity. I read the review by the one reviewer with more familiarity. They did a good job, but I was not convinced by their questions + your responses that the paper received adequate scrutiny in the review process.
> >
> > **So, I have decided to maintain my score, to highlight for the AC this issue. That said, there's every chance that this paper is worthy of acceptance.**
> >
> > Hopefully the AC is more expert in the field, can read the paper, and make the correct decision. I am genuinely sorry to not feel able to raise my score; your work is well thought out and put together, and your responses to the reviews suggest you will make necessary adjustments if accepted. You clearly have a chance (as the other reviews were more positive than me), so it'll be up to the AC -- best of luck.

---

### Official Review · Reviewer_w465 · 2025-03-14

**Overall Recommendation:** 3

**Summary:**

This paper examines fraud-proof mechanisms in subscription platforms. Specifically, the authors define a set of axioms covering fundamental properties, protection against strategic manipulation, and fairness in revenue division mechanisms. They analyze commonly adopted mechanisms and verify which axioms they satisfy. Based on these observations, the authors propose a new mechanism, ScaledUserProp, designed to enhance fairness in revenue distribution by considering the maximum envy metric. Experiments on both real-world and synthetic datasets demonstrate the effectiveness of the proposed mechanism.

---

I remain my score unchanged after rebuttal.

**Claims And Evidence:**

**(Pro)** The theoretical results are generally sound.

**(Con)** In the axioms of fraud-proofness, the final constraint is given as $\phi_{I'}(\hat{C}) - \phi_{I}(\hat{C}) \le \hat{n}$. Why is the right-hand side not generalized to a more flexible form, such as $\hat{n} \cdot A$, where $A$ is a constant? The current formulation seems restrictive to the specific case of $\hat{n}$. Would the theoretical results still hold under this more general setting? A similar concern applies to the bribery-proofness condition.

**Essential References Not Discussed:**

No essential references appear to be missing.

**Experimental Designs Or Analyses:**

**(Con)** In the experiments, the authors analyze the top and bottom few agents based on their pay-per-stream values. However, in Section 4, the theoretical analysis focuses on only the top and bottom single agents. Could the authors clarify why these two evaluation methods differ?

**Methods And Evaluation Criteria:**

The evaluation metrics and methods are generally sound.

**Other Comments Or Suggestions:**

The paper is generally well-written. However, the axioms are quite complex and require significant effort to understand. It would be helpful if the authors included more illustrative examples to clarify the axioms.

**Other Strengths And Weaknesses:**

No additional strengths or weaknesses were identified.

**Questions For Authors:**

See the concerns listed above.

**Relation To Broader Scientific Literature:**

**(Pro)** The paper's key contribution is the comprehensive analysis of existing revenue division mechanisms across a broad range of axioms and the introduction of a new mechanism to improve fairness.

**Theoretical Claims:**

**(Pro)** I did not verify the theoretical claims in full detail, but they appear to be generally sound.

**(Con)** Some claims require further clarification. In Definition 2.4, does fraud-proofness hold for all $\hat{N} \subseteq N$ and $\hat{C} \subseteq C$? I noticed that in Definition 2.6, the condition "for all $\hat{C} \subseteq C$" is explicitly stated. Additionally, in Definition 2.12, should there be a further requirement that $\delta < w\_{ij} - w'\_{ij}$? This would prevent cases where $w'\_{ij} = w\_{ij} - \delta$ is so small that $|w'\_{ij} - w'\_{i'j}|$ becomes larger than $|w\_{ij} - w\_{i'j}|$.

---

> ### Author Rebuttal · Authors · 2025-03-31
>
> Thank you for your review. Please find our responses below.
>
> ---
> > “In the axioms of fraud-proofness, the final constraint is given as $\phi_{I'}(\hat{C}) - \phi_{I}(\hat{C}) \leq \hat{n}$. Why is the right-hand side not generalized to a more flexible form, such as $\hat{n} \cdot A$, where $A$ is a constant? The current formulation seems restrictive to the specific case of $\hat{n}$. Would the theoretical results still hold under this more general setting? A similar concern applies to the bribery-proofness condition.”
>
> The $\hat{n}$ term in both the fraud-proofness and bribery-proofness axioms is a result of the assumption where each user's subscription cost is set to 1, but this is without loss of generality. This assumption implies that creating $\hat{n}$ fake users incurs a total cost of $\hat{n}$. However, the formulation is indeed flexible: if we assume that the cost of creating a user is instead a constant $A$, the right-hand side can be generalized to $\hat{n} \cdot A$ without affecting any theoretical result. Note that our model can also be generalized to consider users with variable cost (refer to our response to Reviewer Qvr6).
>
> ---
> > “Some claims require further clarification. In Definition 2.4, does fraud-proofness hold for all $\hat{N} \subseteq N$ and $\hat{C} \subseteq C$? I noticed that in Definition 2.6, the condition "for all $\hat{C} \subseteq C$" is explicitly stated. Additionally, in Definition 2.12, should there be a further requirement that $\delta<w_{ij}-w'_{ij}$? ...”
>
> For Definition 2.4: Yes that’s right, we will add $\hat{C} \subseteq C$ (the part on $\hat{N} \subseteq N$ is already handled by the universal quantification over instances), thanks!
>
> For Definition 2.12, we noticed a typo in condition (ii), specifically in the second inequality: it should read $w_{i'j}' \leq w_{ij}'$ instead of $w_{i'j} \leq w_{ij}$. This correction ensures that the new engagement profile is “at least as balanced” as before for candidate $j$, aligning with the intended interpretation you mentioned. The proofs remain valid after correcting this typo. Thanks for pointing this out!
>
> ---
> > “In the experiments, the authors analyze the top and bottom few agents based on their pay-per-stream values. However, in Section 4, the theoretical analysis focuses on only the top and bottom single agents. Could the authors clarify why these two evaluation methods differ?”
>
> Our theoretical results on the top and bottom single agents easily extend to top-$k$ and bottom-$k$ agents as well, making it consistent with that used for the experiments. For all our counterexamples, we can simply duplicate each agent $k$ times. We will clarify this point in the revised version.
>
> In our experiments, we chose to report metrics for the top and bottom few agents rather than just the single best and worst, as we believe this provides a more robust assessment—mitigating the impact of potential outliers that may disproportionately affect the extremes.
>
> ---
> We have noted your other comments/suggestions and will take them into account in the revision. Thanks!

---

> > ### Comment · Reviewer_w465 · 2025-04-02
> >
> > Thank you for the response. After considering the other reviews, I have decided to keep my score unchanged.

---

### Official Review · Reviewer_Qvr6 · 2025-03-15

**Overall Recommendation:** 3

**Summary:**

The paper explores fraud-proof revenue division on subscription platforms like Spotify and Apple Music, where users pay a fixed fee for unlimited access, and creators are compensated based on engagement. Current revenue-sharing rules, like GLOBALPROP (proportional to total streams), are vulnerable to manipulation through bots and click farms, making fraud detection complex and computationally hard.

The authors propose three new fraud-resistance axioms—fraud-proofness (preventing profit from fake users), bribery-proofness (preventing profit from bribed users), and Sybil-proofness (preventing profit from splitting/merging identities). Existing rules like USERPROP and USEREQ offer some protection but fail on fairness or manipulation resistance. The paper introduces SCALEDUSERPROP, a new mechanism that adjusts user contributions based on engagement intensity, making it fraud-proof, bribery-proof, and Sybil-proof.

**Claims And Evidence:**

**Supported Claims:**

1- The paper shows that GLOBALPROP fails the fraud-proofness and bribery-proofness axioms.
It also shows that detecting fraudulent activity under GLOBALPROP is computationally NP-hard.

2- USERPROP and USEREQ improve on GLOBALPROP by satisfying fraud-proofness and bribery-proofness.

3- The paper shows that SCALEDUSERPROP satisfies fraud-proofness, bribery-proofness, Sybil-proofness, and fairness axioms (engagement monotonicity and Pigou-Dalton consistency).

**Potentially Problematic Claims:**
The claim that **"SCALEDUSERPROP is the fairest alternative"** is not fully substantiated. While the empirical results support that SCALEDUSERPROP reduces disparity, fairness is inherently subjective and context-dependent. Reducing disparity is not necessarily a positive outcome, as content providers may differ in quality, and the platform may want to incentivize and attract higher-quality artists.

The paper does not provide a **concrete definition of fairness** beyond the Pigou-Dalton consistency and engagement monotonicity axioms. A more thorough discussion of trade-offs—such as how fairness is balanced against platform revenue, user satisfaction, and content quality—would strengthen this claim. Additionally, the paper does not address how the proposed fairness notion might impact the platform’s long-term ecosystem, including its ability to attract and retain high-quality content providers.

**"Detecting fraud under GLOBALPROP is computationally intractable."**
The paper proves that finding the set of artists who benefit the most from fraud is NP-hard.
However, it does not explore whether heuristic or approximate methods could still be effective in practice. Acknowledging this would make the claim more balanced.

**"USERPROP and USEREQ are fraud-proof and bribery-proof."**
The proofs assume that the cost of creating fake users is normalized to 1 unit per user — but in practice, this cost could vary depending on the platform's structure. A sensitivity analysis or discussion of how varying costs would affect these conclusions would strengthen this claim.

**Essential References Not Discussed:**

The paper "Learning Product Rankings Robust to Fake Users" (https://dl.acm.org/doi/10.1145/3465456.3467580) is highly relevant to this study. It focuses on designing learning algorithms for ranking products on digital platforms while being resilient to fake clicks. It would be beneficial for the authors to broaden the discussion on manipulation and mitigation strategies beyond the subscription-based model, considering a more general framework that could apply to various types of digital platforms.

**Experimental Designs Or Analyses:**

See above

**Methods And Evaluation Criteria:**

The paper presents formal proofs that SCALEDUSERPROP satisfies the proposed axioms.

The computational complexity results (e.g., NP-hardness of fraud detection under GLOBALPROP) are well-supported by theoretical analysis.

The evaluation uses both real-world data (Music Listening Histories Dataset) and synthetic data to test the proposed mechanisms.

**Other Comments Or Suggestions:**

1- In Definition 2.4, please clarify what $\hat n$ and $\hat C$ are. And explain better in words, what this definition aims to say. Overall, definitions would benefit from more explicit discussions.

For example, in Definition2.4 (**Fraud-proofness**),  here is how I interepret it. A rule $\phi$ is fraud-proof if an attacker cannot create fake users and profit from them. Here,  $N$ = Set of real users, $\tilde{N} \subseteq N$ = Set of fake users .  $C$ = Set of real artists, $\tilde{C} \subseteq C$ = Set of fake artists. $w_i$ = Engagement profile of user $i$.
Fraud-proofness holds if the extra profit from fake users does not exceed the cost of creating them:
$
\phi_{I'}(\tilde{C}) - \phi_I(\tilde{C}) \leq |\tilde{N}|
$
where $|\tilde{N}|$ is the number of fake users (cost is assumed to be 1 unit per user).

For **Bribery-Proofness**, here is my understanding:  Bribery-proofness means that an attacker cannot profit by bribing real users to increase engagement with specific artists.
Cosnider  any two instances:   $I = (N, C, w)$ = Initial setup
and  $I' = (N, C, w')$ = Setup after changing the engagement of exactly $k$ users  e
The rule is bribery-proof if:  $
\phi_{I'}(\tilde{C}) - \phi_I(\tilde{C}) \leq k
$
where:
 $\phi_{I'}(\tilde{C})$ = Revenue to attacker’s artists after bribing;
$\phi_I(\tilde{C})$ = Revenue to attacker’s artists before bribing; and
$k$ = Number of bribed users (cost assumed to be 1 unit per user)



2- Please recall the definition of $\alpha$ in Theorem 2.8

3- Also, try to better explain in words the GlobalProp and UserProp. Here is my understanding: GLOBALPROP (Global Proportional Rule) works as follows: (i) Revenue is pooled into a global pot. (ii) Each artist’s payment is proportional to their share of the total engagement across the entire platform. USERPROP (User Proportional Rule) works as follows:
(i) Each user’s subscription fee is treated as an individual pool. (ii) Each artist’s payment is based on their share of that specific user’s engagement. (iii) Revenue is distributed based on what individual users actually listen to, rather than global totals.

**Other Strengths And Weaknesses:**

The exposition could be improved in several places, as parts of the paper appear rushed or underdeveloped.

I strongly recommend that the authors improve the positioning of this work within the broader literature. Adding a dedicated related work section, even if placed in the appendix, would help clarify how this work builds on and differs from existing research.

**Questions For Authors:**

See above

**Relation To Broader Scientific Literature:**

This paper contributes to the broader literature on robustness to strategic manipulations in digital platforms. Similar issues have been studied in the context of online advertising markets. Notable examples include

"Dynamic Incentive-Aware Learning: Robust Pricing in Contextual Auctions" (http://papers.neurips.cc/paper/9169-dynamic-incentive-aware-learning-robust-pricing-in-contextual-auctions.pdf) and

"Dynamic Reserve Prices for Repeated Auctions: Learning from Bids" (https://link.springer.com/chapter/10.1007/978-3-319-13129-0_17).

Additionally, there are empirical studies on fake reviews, such as "An Empirical Investigation of Online Review Manipulation" (https://www.aeaweb.org/articles?id=10.1257/aer.104.8.2421) and "The Market for Fake Reviews" (https://pubsonline.informs.org/doi/10.1287/mksc.2022.1353).

Also, the paper contributes to the literature on fairness in recommendation systems. Example include
Interpolating Item and User Fairness in Multi-Sided Recommendations, (https://nips.cc/virtual/2024/poster/93355)
User-item fairness tradeoffs in recommendations (https://openreview.net/pdf?id=ZOZjMs3JTs)

**Theoretical Claims:**

See above

---

> ### Author Rebuttal · Authors · 2025-03-31
>
> Thank you for your review. Please find our responses below.
>
> ---
> > The claim that “SCALEDUSERPROP (SUP) is the fairest alternative” is not fully substantiated. ... Additionally, the paper does not address how the proposed fairness notion might impact the platform’s long-term ecosystem, including its ability to attract and retain high-quality content providers.
>
> We note that a central focus of our paper is on manipulation-resistance, as emphasized in the title, abstract, and throughout the main text. Fairness is a desirable property, but it plays a complementary role in our work. Broader questions about how fairness interacts with platform objectives and long-term ecosystem health are important, but fall outside the scope of our paper.
>
> That said, we are careful in our claim, stating that “on real-world data, SUP emerges as the fairest mechanism among those considered”. To substantiate this claim empirically, we consider the disparity in the value of each users’ stream (“pay per stream”)—that is, the extent to which streams from different users are valued unequally. Prior work (e.g., Dimont, 2018; Meyn et al., 2023; refs in main text) has highlighted that high disparities in per-stream payments can plausibly be viewed as unfair, particularly when artists are paid unequally for equivalent listener engagement. Such disparity under USERPROP (UP) arises even for a single artist receiving the same number of streams from different users—making it unrelated to artist quality and thus, we argue, difficult to justify from a fairness standpoint.
>
> While rewarding “high-quality” artists is a valid platform design choice, defining “quality” is normative and beyond our scope. In this work, we evaluate fairness conditional on observed engagement, and within this scope, we argue that SUP provides a more defensible approach by “treating equal engagement more equally”. If one equates quality with engagement, our experiments show that SUP better rewards high-engagement artists compared to UP.
>
> ---
> > The paper proves that finding the set of artists who benefit the most from fraud is NP-hard. However, it does not explore whether heuristic or approximate methods could still be effective in practice. Acknowledging this would make the claim more balanced.
>
> We note that the problem we reduce from (SSBVE) cannot be approximated better than $O(n^{1/4})$ under plausible complexity conjectures. With our reduction, for a threshold $\gamma$, under the same complexity conjecture, this rules out any constant factor approximation of the size $k$ such that a set of $k$ artists have _potentially suspicious profit_ exceeding the threshold $\gamma$.
>
> That said, heuristics may work well in practice, and certain “gap versions” (e.g., distinguishing no manipulation from large manipulation) could be polynomial-time solvable. Exploring such variants would be an interesting direction for future work.
>
> ---
> > “"USERPROP (UP) and USEREQ (UEQ) are FP and BP" The proofs assume the cost of creating fake users is normalized to 1 unit per user—but in practice, this cost could vary depending on the platform's structure...”
>
> We note that UP and UEQ could be extended with varying cost while still maintaining fraud-proofness (FP) and bribery-proofness (BP). If we let the cost a user pay to be $c_i$, UP could be defined as $\phi_{I}(j) = \sum_{i \in N} \frac{w_{ij}c_i}{\sum_{j' \in C} w_{ij'}} \times \alpha$ and UEQ could be defined as $\phi_{I}(j) = \sum_{i \in N} \frac{\mathbf{1}[{w_{ij}>0}]c_i}{| set(j' \in C:w_{ij'}>0)|} \times \alpha$. The defns of FP and BP would need to be adapted accordingly.
>
> As this is the first work in this direction, we chose to focus on a simpler model to introduce our approach. We agree though, that with varying costs, there is a richer space for mechanisms that we have not explored, and would make for interesting future work.
>
> ---
> > “beneficial for the authors to broaden the discussion...beyond the subscription-based model, considering a more general framework that could apply to various types of digital platforms.”
>
> Our focus is on the subscription-based model, which is already a rich and nuanced setting with many compelling research questions in its own right. That said, we agree that extending the framework to encompass a broader class of digital platforms would be a valuable and interesting direction for future work.
>
> ---
> > “I strongly recommend that the authors improve the positioning of this work within the broader literature. Adding a dedicated related work section, even if placed in the appendix, would help clarify how this work builds on and differs from existing research.”
>
> We will expand the current related work section to better situate our contributions within the broader literature on robustness to strategic manipulations in digital platforms (including online advertising markets). Thanks for the helpful references.
>
> ---
> We have noted your other comments/suggestions and will take them into account in the revision.

---

### Decision · Program_Chairs · 2025-05-01

**Decision:**

Accept (poster)

**Comment:**

This paper studies how to design equitable and fraud-resistant revenue-sharing mechanisms for streaming platforms with subscribe-based nature. The authors show that existing approaches are vulnerable to various types of frauds. Motivated by this, the authors propose three new axioms, including fraud-, bribery- and Sybil-proofness, and then introduce a new mechanism “ScaledUserProp”. They show that this mechanism meets all of those three axioms and certain fairness criteria simultaneously. The authors validate the effectiveness of their mechanism with theoretical results and numerical experiments (on synthetic data and real-world scenarios).

This work is clearly motivated, smoothly delivered and rigorously proved. Their mechanism, also serving as a methodology, resolve the real problem that manipulation could undermine standard revenue-sharing methods. Although some of the reviewers are suggesting more experiments, given the theoretic nature of this work, I still consider the overall quality of this work reaches the criteria of acceptance.

The only remaining concern lies in the relevance to ICML. From my own understandings, mechanism design and game theory are categories of ICML submission, and this paper proposes a mechanism which is also a methodology of revenue sharing and fraud resisting. Given that they are proposing methods of advancing ML, I think it is proper for us to accept it here.